# Micro and macroevolution of sea anemone venom phenotype

Edward G. Smith [1,5,7] ✉, Joachim M. Surm [2,7] ✉, Jason Macrander[1,3], Adi Simhi[2,4], Guy Amir[2,4], Maria Y. Sachkova[2,6], Magda Lewandowska[2], Adam M. Reitzel [1] & Yehu Moran [2] ✉

Venom is a complex trait with substantial inter- and intraspecific variability resulting from strong selective pressures acting on the expression of many toxic proteins. However, understanding the processes underlying toxin expression dynamics that determine the venom phenotype remains unresolved. By interspecific comparisons we reveal that toxin expression in sea anemones evolves rapidly and that in each species different toxin family dictates the venom phenotype by massive gene duplication events. In-depth analysis of the sea anemone, *Nematostella vectensis*, revealed striking variation of the dominant toxin (*Nv1*) diploid copy number across populations (1-24 copies) resulting from independent expansion/contraction events, which generate distinct haplotypes. *Nv1* copy number correlates with expression at both the transcript and protein levels with one population having a near-complete loss of Nv1 production. Finally, we establish the dominant toxin hypothesis which incorporates observations in other venomous lineages that animals have convergently evolved a similar strategy in shaping their venom.

Understanding the molecular processes that drive phenotypic diversity among species, populations, and individuals is essential for unraveling the link between micro and macroevolution. Most traits are polygenic, meaning that their phenotype is influenced by multiple genomic loci[1–5]. However, understanding the heritability of these complex traits is challenging. Gene expression is likely an essential feature in determining the type of effect a gene has on a polygenic trait. This is evident with heritable gene expression dynamics contributing to phenotypic variations within and between species[6,7]. The mechanisms that drive these gene expression dynamics, which include mutations to the cis- and trans-regulatory elements[8,9], epigenetic modifications[7,10,11], and gene duplication[12–14], are subject to selective pressures that can result in adaptive traits in an organism.

Among the mechanisms capable of driving rapid shifts in gene expression dynamics is gene duplication, which can cause an increase in transcript abundance leading to phenotypic variations within and between species. Gene duplications, resulting in copy number variation (CNV), can originate from a combination of replication slippage, unequal crossing over during meiosis, retroposition of gene transcripts, and whole-genome duplications[15,16]. In addition to providing substrate for molecular evolution to act on via diversification, CNV arising from gene duplications can also cause immediate fitness effects resulting from increased gene expression through dosage[17]. Indeed, the potential for immediate phenotypic effects and the high mutation rates of duplicated genes suggest that CNV may be an important mechanism for rapid adaptation to new ecological niches. While CNV is studied mostly in the context of human genetic diseases and recent

[1]University of North Carolina at Charlotte, Department of Biological Sciences, Charlotte, NC, USA. [2]Department of Ecology, Evolution and Behavior, Alexander Silberman Institute of Life Sciences, Faculty of Science, The Hebrew University of Jerusalem, Jerusalem, Israel. [3]Florida Southern College, Biology Department, Lakeland, FL, USA. [4]The Hebrew University of Jerusalem, The School of Computer Science & Engineering, Jerusalem, Israel. [5]Present address: School of Life Sciences, University of Warwick, Coventry, United Kingdom. [6]Present address: Sars International Centre for Marine Molecular Biology, University of Bergen, Bergen, Norway. [7]These authors contributed equally: Edward G. Smith, Joachim M. Surm. ✉e-mail: ed.g.smith@warwick.ac.uk; joachim.surm@mail.huji.ac.il; yehu.moran@mail.huji.ac.il

adaptations[18–20], there is an increasing appreciation for the role of individual and population-scale CNV in ecological and evolutionary processes in other species and its impact on complex traits[21–23].

A complex trait hypothesized to be evolving under strong selective pressure is venom due to its essential ecological roles related to predation and defense[24–26]. The venom phenotype is often a complex trait because it relies on the coordinated expression of multiple toxin-coding genes. These toxins combine to produce venom profiles that are highly distinct, varying significantly within and between species[26,27]. Evidence supports that differences in toxin gene expression among species are a major contributor to the rapid evolution of venom phenotypes[28,29]. Toxin gene families have been hypothesized to evolve through a birth and death model of adaptive evolution as these proteins are central to an individual's fitness in mediating the interactions for both nutrition and survival[24,25]. Comparative genomics has revealed evidence supporting a hypothesis that a number of venomous organisms rapidly accumulate gene duplications in their genomes: examples include spiders[30,31], cone snails[32], and scorpions[33], although there are exceptions such as widow spiders[34].

Cnidarians represent an ancient venomous phylum where likely all species rely on toxins for prey capture and defense from predators[35]. Among cnidarians, sea anemone venom is arguably the most well-characterized[36] and past research has shown that toxin gene duplication is an important feature in these organisms[13,37,38]. Various sea anemone toxin families have been structurally and functionally validated or their expression localized to epithelial gland cells and specialized stinging cells called nematocytes[39,40]. These include pore-forming toxins such as Actinoporins[41,42], neurotoxins such as Nematocyte Expressed Protein 3 (NEP3[43,44]), sodium channel modulators (NaTx[45,46]), potassium channel toxins (KTx type 1, 2, 3, and 5 families[36,43,47–49]) and proteases such as NEP6 Astacins[44]. The characterization of these venom components has led to the investigation of their phylogenetic and evolution histories, revealing that these toxin families evolve under purifying selection[37,42,50], with the exception of KTx3 which has been shown to evolve under the influence of diversifying selection[50]. One of the most well-characterized cnidarian toxins is the *Nv1* family from the estuarine sea anemone, *Nematostella vectensis* Stephenson, 1935. Located in the ectodermal gland cells[40], this sodium channel toxin is the major component of the *N. vectensis* venom and has previously been shown to be encoded by at least 11 nearly identical genes that are clustered on one chromosome[13,51,52]. Furthermore, population-specific variants of *Nv1* absent from the reference genome assembly have been identified at specific locations across this species' geographic range along the Atlantic coast of the United States[13] and suggests the potential for location-specific alleles and the presence of unresolved intraspecific variability in the *Nv1* gene family.

Here, we investigate the evolution of venom in sea anemones at both macro- and microevolutionary scales. We employed a combination of comparative transcriptomics and modeling to understand the macroevolution of venom as a complex trait in sea anemones to reveal that toxin expression evolves rapidly among sea anemones with little constraint in their combinations. We find that in sea anemones, a single toxin family dominates their venom phenotype and can dynamically shift even between closely-related species or convergently evolve among distantly-related species. Phylogenomic analysis supports that the dominant toxin family undergoes massive gene duplication events. By investigating different populations of *N. vectensis* using a combination of transcriptomics, long-read genome sequencing, genomic qPCR, and proteomics, we further show that significant expansion and contractions events are driving dynamic shifts in the gene expression of the dominant toxin even at the microscale.

## Results

### Macroevolution of sea anemone venom phenotype

To investigate the macroevolution of venom as a complex trait, we employed comparative transcriptomics to quantify the gene expression of different toxin components and generate the venom expression phenotype among sea anemone species. Using publicly available transcriptomes, we identified single-copy orthologs to reconstruct the relatedness among sea anemones (Fig. 1A and Supplementary Data 1). In concert, we mapped the expression of multiple toxin families to each de novo assembled transcriptome. This included Actinoporin, NEP3 and NEP6, NaTx, and KTx1, 2, 3, and 5. Transcripts per million (TPM) values generated from the mapping were then used to reconstruct the venom expression phenotype for each species (Fig. 1A, pie graphs at tips). By performing ancestral state reconstruction (ASR) of the venom expression phenotype among sea anemones (Fig. 1A), we revealed that the NaTx toxin family was most likely the dominant toxin in the last common ancestors of sea anemones.

For most sea anemones (17 of 29), a single toxin family contributed to the majority of the venom expression phenotype and accounted for >50% of the total toxin expression (Supplementary Data 2). During diversification of Actinioidea, ASR suggests that KTx3 evolved to become the dominant toxin family. The KTx3 family is the dominant toxin family in 10 of the 17 Actinioidea species, with Actinoporin, KTx1, and KTx2 dominant in four, one and two species, respectively. Outside of Actinioidea, the Edwardsiid *Scolanthus callimorphus* Gosse, 1853 convergently evolved to have KTx3 as the dominant toxin. These shifts in the dominant toxin can be explained by a model of punctuated evolution[53,54]. We tested this by modeling the rates of evolution acting on the expression of toxins. We find evidence that all sea anemone venom components undergo dramatic and unique shifts that is best explained through a mode of rapid pulses (Pulsed) as opposed to Brownian motion (BM), Ornstein–Uhlenbeck (OU), or early burst (EB) models (Fig. 1C).

To understand the constraint acting on the toxin families themselves as well as the combinations of toxins they can form, we performed phylogenetic covariance analysis. Broadly, our analysis shows that sea anemones have minimal constraint acting on the combinations of toxins they employ to capture prey and defend against predators (Fig. 2A and Supplementary Data 3). While our results revealed that the venom expression phenotype of sea anemones has considerable flexibility in the combinations of toxins they express, there was an exception with NEP3 neurotoxin[43], and NEP6 protease families[44], which have a significant correlation in their expression. In concert, these two toxin families have the most pronounced phylogenetic signal in their expression among all toxins (Supplementary Data 4, with a strong signal having values close to 1), providing evidence that the expression for each toxin family is more similar among closely related species.

We then explored the venom expression phenotype of sea anemones by clustering the phylogenetic covariance of toxin expression using principal component analysis (PCA; Fig. 2B). This reconstructed the phylomorphospace of sea anemone venom, revealing that this complex trait has relatively low dimensionality (Supplementary Fig. 1), with two principal components accounting for the majority of variation (62%). While our analysis focused on transcriptomes generated from adults, RNA was generated from different tissue types with the majority coming from multiple tissue types. We therefore tested whether different tissues impacted this our results by using tissue type as a fixed effect in our PCOV analysis and found that this was not significant (Supplementary Data 5). Broadly, the venom expression phenotype clustered together depending on the toxin family with the highest expression, even among distantly-related species found in different superfamilies. While the expression of NEP3 and NEP6 show significant phylogenetic covariance, this had little impact on the broad clustering of the venom expression phenotype among sea anemones.

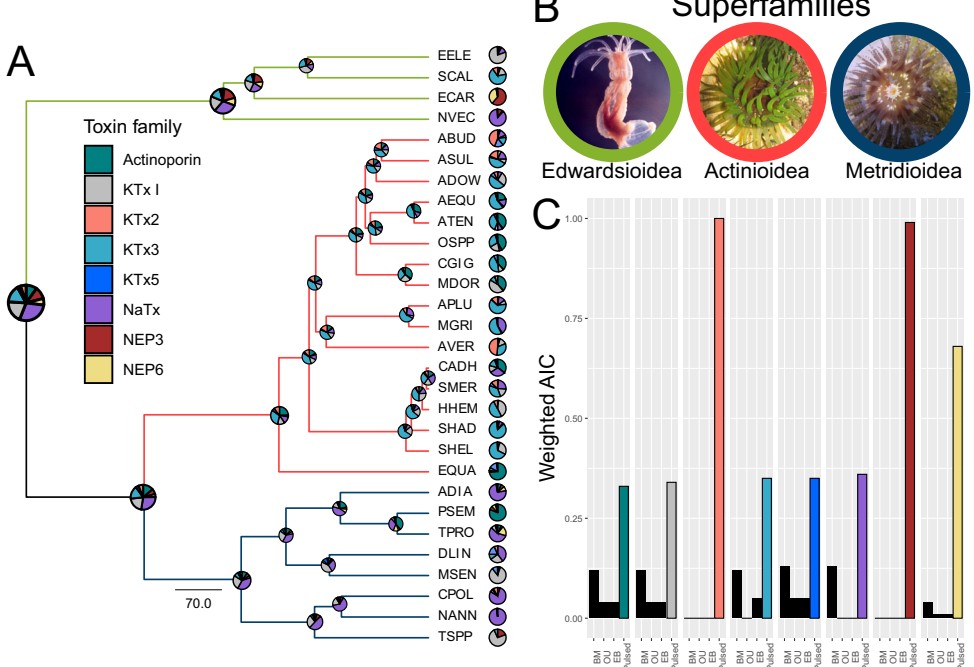

**Fig. 1 | Ancestral reconstruction and phylogenetic constraints on the venom expression phenotype in Actiniaria. A** Phylogenetic tree of sea anemones with pie charts nodes represents the ancestral reconstruction of known toxins and pie chart at tips represents the venom expression phenotype. All nodes had ultrafast boot-strap support >95% at nodes. **B** Representative images of the three sea anemone superfamilies included in the phylotranscriptomic analyses (Edwardsioidea−*N.*

*vectensis;* Actinioidea−*Aulactinia veratra;* Metridioidea−*Calliactis polypus*). Acti-nioidea and Metridioidea photos courtesy of Peter Prentis. **C** Models of trait evo-lution fitted to toxin expression highlights that pulsed evolutionary process best describes sea anemone venom evolution. Model of best fit highlighted in color based on weighted AIC and are colored according to the toxin family key in panel **A**. See Supplementary Data 1 for species code and reference.

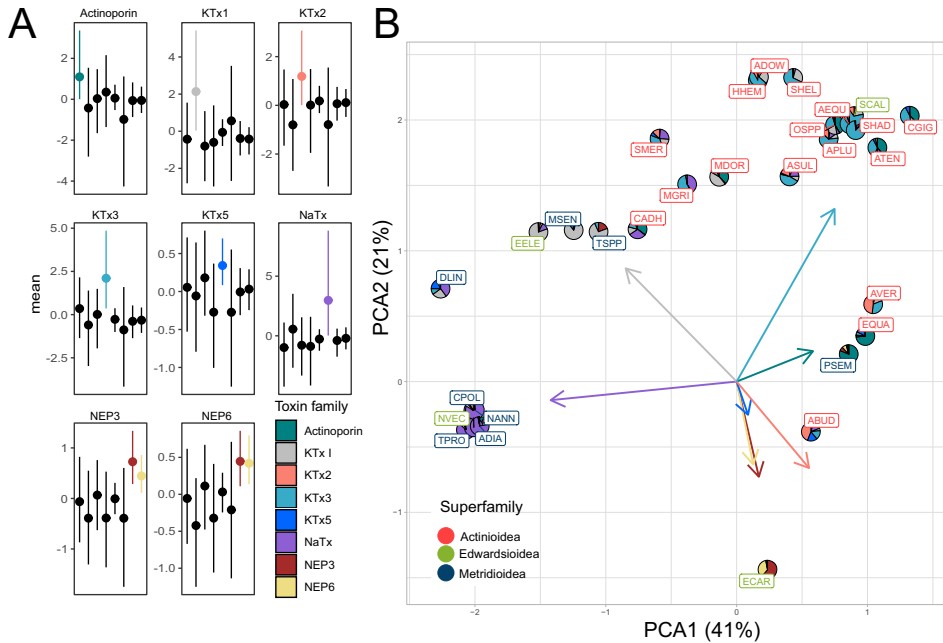

**Fig. 2 | Sea anemone venom expression phenotype characterized by a single dominant toxin that evolves through major pulses. A** Toxin combinations with significant phylogenetic covariance represented in color, with those not significant in black and lower and upper 95% confidence intervals represented as whiskers.

Phylogenetic covariance analysis was run using Markov chain Monte Carlo for a total of 20 million iterations. **B** Sea anemone venom phylomorphospace cluster on the toxin family that contributes to the majority of expression. Loadings for each toxin family represented as arrows.

Furthermore, KTx3 and NaTx are the two toxin families that have the largest loadings, suggesting that they are the major families that define the venom profile employed by sea anemones. These findings suggest that in sea anemones a single dominant toxin family is the major driver in dictating the venom expression phenotype of each species.

Taken altogether, these results highlight that although the venom in sea anemones is comprised of many different toxins, we see that a single toxin family dominates the venom expression phenotype in sea anemones. This is supported by evidence that for most toxins, the phylogenetic signal acting on toxin expression is weak, the venom expression phenotype has low dimensionality, and that little constraint appears to be acting on the combinations of toxins expressed. Furthermore, the evolution of toxin expression appears to be highly dynamic, undergoing a process of rapid pulses. Strikingly, we see convergent shifts in the venom expression phenotype among distantly related species, likely the result of independently adapting the same dominant toxin family.

## Genomic architecture of the dominant toxin family

While our comparative transcriptomics and phylogenetic covariance analysis revealed that the sea anemone venom expression phenotype is largely dictated by a single toxin family, the genetic architecture that underlies the dominant toxin family requires investigation at the genomic level. We investigated the sea anemone genomes recently assembled using long-read sequencing for three species, *Actinia equina* (Linnaeus, 1758)[55], *S. callimorphus*, and *N. vectensis*, from two superfamilies (Actinioidea and Edwardsioidea). Remarkably, we find evidence that massive duplication events underly the signal driving a toxin family to become dominant (Fig. 3A), with all three sea anemone species possessing more than 15 copies of each of their respective dominant toxin gene family. Our phylogenetic covariance analysis and comparative transcriptomics revealed that in *S. callimorphus* and *A. equina*, the dominant toxin is KTx3, whereas, in *N. vectensis*, the dominant toxin is NaTx. For each species, the dominant toxin family accounts for the highest number of copies among all other toxin families (Supplementary Data 6). While *A. equina* contains both NaTx and KTx3, the KTx3 toxin family underwent a much greater series of duplication events, with eight members from the NaTx family and 52 members from the KTx3 family.

Next, we aimed to unravel the evolutionary steps that led to the amplification of the dominant toxin family in sea anemones by investigating the genomic location and macrosystemic relationship of chromosomes/scaffolds. To do this we performed phylogenomic analyses and discovered that macrosynteny is broadly shared among the three species (Fig. 3B), which confirms that the macrosyntenic relationship of chromosomes between *N. vectensis* and *S. callimorphus* is consistent with previous analyses[56]. This is particularly evident between *N. vectensis* and *S. callimorphus* whose assemblies utilized long-read sequencing and high-throughput chromosome conformation capture to generate chromosome-level genome assemblies, whereas *A. equina* genome was generated from only long-read sequencing. From our analysis, we find 15 chromosomes are linked between *N. vectensis* and *S. callimorphus*, and that these are linked to 108 scaffolds found in *A. equina* (Fig. 3B and Supplementary Data 7). Our analysis further reveals that while macrosynteny is largely conserved among the three species, synteny among toxin loci for the KTx3 family in *S. callimorphus* and *A. equina*, or the NaTx family in *N. vectensis* and *A. equina*, is absent. This was further confirmed by exploring the genes and genomic sequence up and downstream of each toxin loci. In contrast, the NEP3 and NEP6 gene families can be seen to lie on scaffolds that share macrosynteny among the three genomes (Supplementary Data 8). This supports that the evolution of genes encoding some toxin families are highly dynamic including the NaTx and KTx3 families which have become dominant in *N. vectensis*, and *S. callimorphus* and *A. equina*. Interestingly, while these two toxins are

distinct from each other (Fig. 3C), evident from the CLANS clustering, they likely share a common evolutionary history, which is supported by evidence that they share the same cysteine framework (Fig. 3D) and some KTx3 toxins having similar activity to NaTx toxins[57–60]. Because of this likely shared evolutionary history, we also explored whether any synteny was shared between the NaTx and KTx3 families to test a hypothesis for an ancestral NaTX/KTx3, however, no macro or microsynteny was found. This further suggests that these toxin families undergo rapid evolution in their genomic architecture compared to other genes and even other toxin genes.

We further explored the molecular evolution of the dominant toxin family within each species to gain insight into the modes of gene duplication that might be shared among species. In *N. vectensis*, 14 of a total of 18 NaTx copies share 99% sequence similarity at the mRNA level and were hypothesized to evolve through tandem duplication and possibly concerted evolution to result in the *Nv1* cluster[13]. In *A. equina*, four NaTx copies are found on a single cluster, with another four located throughout the genome, yet still they share an average of 87% similarity at the mRNA level. In *S. callimorphus* and *A. equina*, KTx3 copies frequently also cluster together in tandem, however, they also underwent repeated translocation events. They also do not display the same degree of gene homogenization observed for the *Nv1* cluster or NaTx copies in *A. equina*, with *S. callimorphus* and *A. equina* KTx3 copies sharing 73% and 34% similarity at the mRNA level, respectively (Supplementary Data 9). These results support that the amplification of the KTx3 gene family is likely occurring through lineage-specific duplications, and that tandem duplication events play a major role for both NaTx and KTx3 families.

Overall, our comparative transcriptomics and phylogenomic analysis have provided striking insights into the macroevolution of venom in sea anemones. From these analyses, we see that a dominant toxin family dictates the venom expression phenotypes in sea anemones and that this evolves in a highly dynamic process through rapid pulses that are driven by gene duplication events. However, it is unclear how these patterns occur at the population and individual scale and understanding this link would provide important insights into the microevolution of venom in sea anemones.

## Population dynamics of the venom phenotype in *N. vectensis*

Previous work has revealed that the *N. vectensis* NaTx cluster of genes are overall highly similar but also that population-specific variants of *Nv1* exist[13]. This led us to explore the population dynamics of the *Nv1* cluster in *N. vectensis* by performing comparative transcriptomics, quantitative genomic copy number PCR, proteomics, and genomics using long-read sequencing.

To explore the microevolution of the venom phenotype in *N. vectensis*, we first needed to understand its population structure across the native geographical range along the Atlantic coast of North America. To do this, highly complete transcriptomes were generated from nine *N. vectensis* populations originating from locations on the Atlantic coast of North America (Fig. 4A). Specifically, all transcriptomes had a BUSCO score >90%, except for Massachusetts (Supplementary Data 10, BUSCO = 72.2%). In all, 2589 single-copy orthologs were identified using OrthoFinder and used to generate a well-supported maximum-likelihood phylogenetic tree (Fig. 4B). Broadly, populations clustered according to geographical location, with populations from North (Massachusetts, Maine, New Hampshire, New Jersey, and Nova Scotia) and South (North Carolina, South Carolina, and Florida) clustering independently together. This phylogenetic analysis also supports that the Maryland population, which serves as the source for the most common *N. vectensis* lab strain[61], clusters more closely with southern populations, consistent with the previous analyses[62]. Differences among populations from close geographical locations are also observed, specifically with South Carolina populations clustering more closely with Florida than North Carolina.

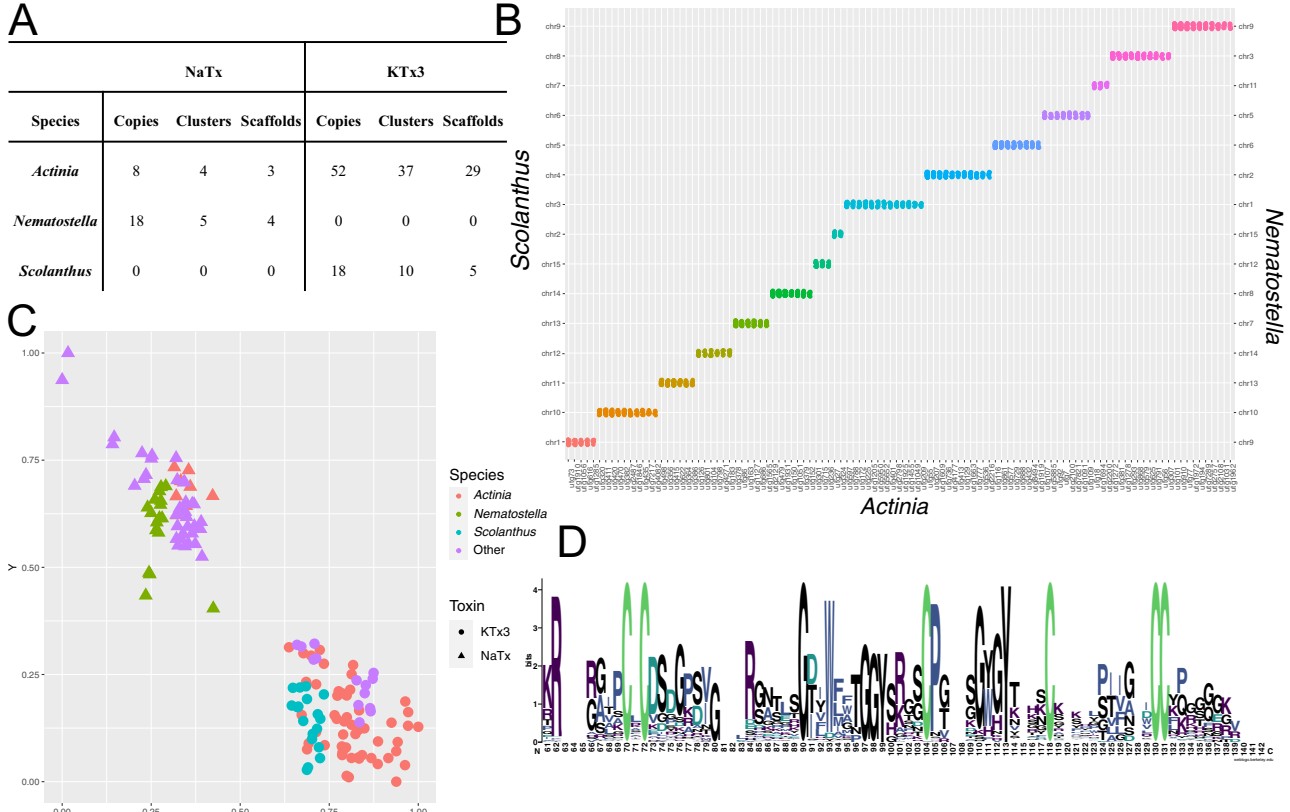

**Fig. 3 | Phylogenomic analysis of the dominant toxin family in *A. equina*, *N. vectensis*, and *S. callimorphus*. A** Table representing the copy number of toxins found across genomic scaffolds assembled. **B** Oxford plot representing the macrosyntenic relationships of chromosomes among *A. equina*, *N. vectensis*, and *S. callimorphus*. **C** Pairwise similarity based clustering of NaTx and KTx3 found in the three genomes as well as other sea anemone species by Cluster Analysis of Sequences (CLANS) software[113]. Sequences from *A. equina*, *N. vectensis*, and *S. callimorphus* represented by different colors. Other includes toxin used from previous work[50]. The KTx3 and NaTx families are represented as circles and triangles, respectively. **D** Shared evolutionary history of NaTx and KTx3 highlighted by a conserved cysteine framework identified in the mature peptide.

After reconstructing the population structure of *N. vectensis*, we then aimed to explore the venom phenotype among populations. We focused on the evolution of the NaTx gene family, which is the dominant toxin family in the model representative *N. vectensis*. Our comparative transcriptomic approach identified allelic variation for the members of the NaTx gene family (Fig. 4A). Specifically, we were able to capture variants of *Nv1* from all populations with more than eight variants captured in all populations, except for Florida samples in which we were only able to capture a single variant. A possible explanation for only a single variant being captured in Florida is this copy is highly conserved and still maintained in high copy numbers. Investigating the expression patterns for *Nv1* among all populations, however, revealed that *Nv1* has massively reduced expression in Florida with TPM for *Nv1* in all populations >500, while Florida had a TPM of five (Supplementary Data 11A). Expression differences of *Nv1* among populations were further validated using nCounter platform, revealing that indeed *Nv1* gene expression is massively reduced in the Florida population (Supplementary Data 11B). This striking result suggests that the *Nv1* cluster in Florida has undergone a massive contraction.

While we were able to get a representation of the sequence diversity of *Nv1* among populations, capturing the copy number variation of *Nv1* is beyond our capacity using comparative transcriptomics. This is especially significant for the *Nv1* family which can contain identical gene copies within the loci. Therefore, we performed individual quantitative PCR estimates of *Nv1* diploid copy number, (Fig. 4C and Supplementary Data 12) for five populations (North Carolina, Maine, Massachusetts, New Hampshire, and Nova Scotia). From this we found that *Nv1* copy number ranges from 8 to 24 genomic copies for different populations on the Atlantic coast of North America. We observed significant differences in the mean copy number across populations (ANOVA, $P < 2e-16$; Supplementary Data 13), with pairwise post hoc tests revealing significant differences among all population pairs (Tukey HSD, $P < 0.05$; Supplementary Data 14), except the Maine-New Hampshire comparison. The mean population copy number was lowest in Maine and New Hampshire (11 copies) and highest in North Carolina (20 copies).

While genomic and transcriptomic measurements can provide the copy number and expression level of a gene, respectively, the biology of a trait heavily depends on the synthesis level of the protein product of a gene. Moreover, in some cases protein levels are not in direct correlation to RNA levels, and proteomic and transcriptomic dataset might give contrasting pictures[63–65]. Thus, we tested the notion that Florida Nv1 protein levels are massively reduced using a proteomics approach, comparing samples from Florida with North Carolina, the closest population to Florida where we had genomic data. This analysis revealed that Nv1 in Florida is at either negligible or undetectable levels, both when using iBAQ and label-free quantification (LFQ) values. In contrast, Nv1 in North Carolina was measured as the third most abundant protein in the whole proteome (Supplementary Data 15), resulting in Nv1 being the most significantly differentially abundant protein between the two populations (Fig. 4D and Supplementary Data 16). This striking difference cannot be explained by a technical limitation in measuring the Florida samples as overall iBAQ and LFQ values were similar for most proteins in the two populations, and the two proteomes significantly correlated ($R^2 = 0.98$; Supplementary Data 17 and Supplementary Fig. 2). Thus, we see a clear

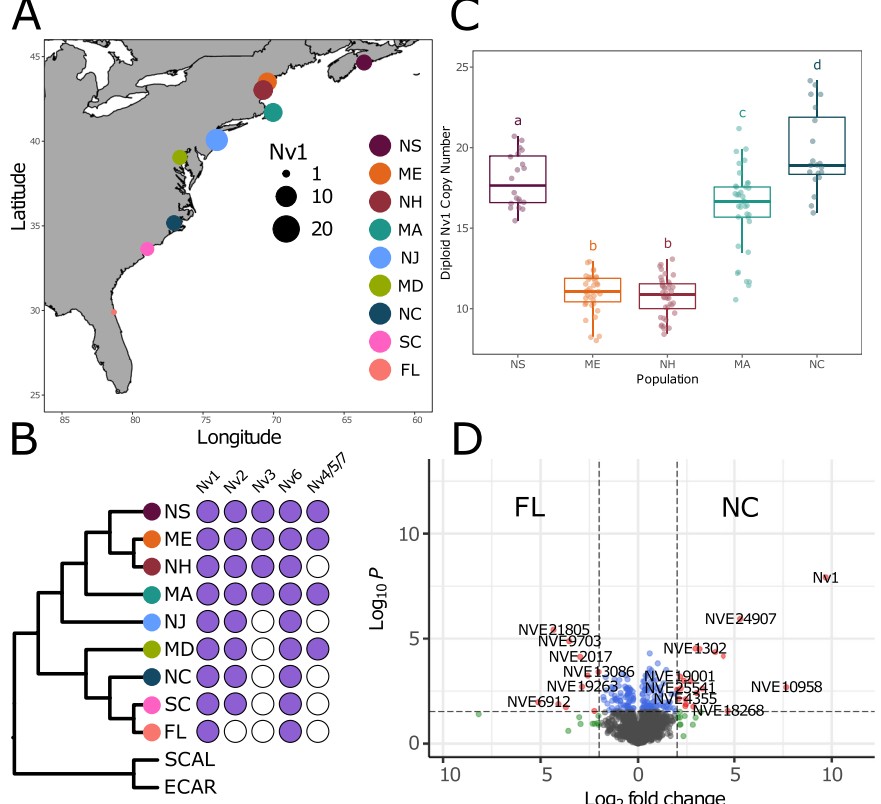

**Fig. 4 | Diversity of NaTx paralogs among *N. vectensis* populations. A** Map showing the location of the sampled populations across North America. Allele diversity represented by size of dot plot at different locations. Florida (FL), Massachusetts (MA), Maryland (MD), Maine (ME), North Carolina (NC), New Hampshire (NH), New Jersey (NJ), Nova Scotia (NS), and South Carolina (SC). **B** Population structure of *N. vectensis* generated using a maximum-likelihood tree from protein sequences and presence/absence of previously characterized NaTx paralogs in *N. vectensis*. **C** Boxplots of diploid copy number estimated using qPCR from samples collected from the five populations. The median is represented by the bold horizontal line and the upper and lower hinges represent the 75th and 25th percentiles, respectively. The boxplot whiskers extend to the largest and smallest values within 1.5 * IQR (interquartile range) for the upper and lower whiskers, respectively. All

individual copy number estimates are shown for each boxplot. ANOVA revealed a significant effect of population on diploid copy number (one-tailed test, $P < 2\text{e-16}$; Supplementary Data 13). The letters above boxplots indicate the results of a Tukey–Kramer post hoc test controlling for the family-wise error rate ($a = 0.05$), with all population comparisons showing significant copy number differences ($P < 0.05$; see Supplementary Data 14 for pairwise comparisons) unless they share the same letter. **D** Volcano plot representing proteins of significantly different abundance, measured as label-free quantification (LFQ) intensity, between FL and NC. Gray dots represent proteins that are not significant, blue dots are proteins with significant $P$ value < 0.01, green dots are proteins with Log2 fold change of >2, red dots are proteins with significant $P$ value and Log2 fold change.

correlation between the reduction in allelic variation found in Florida samples, the reduction in *Nv1* copies, and exceptional reduction of Nv1 at the protein level in the Florida population.

A slight variation in the members of the NaTx gene family were also captured in the transcriptomes generated from different populations. While numbers did vary, all transcriptomes captured at least a single variant of *Nv6*, with some as many as three (Fig. 4B). *Nv2* was captured in all transcriptomes, except for Florida. Notably, *Nv3*, a previously identified variant that contains a 6-bp deletion altering the N-terminus of the mature peptide[13], is located within the *Nv1* locus unlike other more distinct variants (e.g., *Nv4* and *Nv5*) that have translocated outside the *Nv1* locus[66]. Although *Nv3* is not widely identified in our amplicon analyses, this may be influenced by the presence of mutations in the primer binding sites revealed by our genomic analyses. This is likely the case as our comparative transcriptomics was able to recover *Nv3* copies in all North populations (including Massachusetts, Maine, New Hampshire, and Nova Scotia) and absent in all other populations. No copies of *Nv8* were captured, which is also consistent with previous finding that this member of the NaTx gene family is expressed at very low levels. The distribution of *Nv4*, *Nv5*, and *Nv7* is patchy, which may be explained by their expression being restricted to early life stages and maternally deposited in the egg and hence only

captured if individuals sampled were females containing egg packages[66].

Here, we provide multiple lines of evidence that confirms that the copy number of *Nv1*, a member of the NaTx family that is the dominant toxin in *N. vectensis*, evolves in a highly dynamic manner among populations, while other toxin families appear to be much more stable. This is most striking in the Florida population that has undergone a dramatic contraction of the *Nv1* cluster, resulting in the almost total loss of *Nv1* at the mRNA and protein level. This highlights that even within the dominant toxin family (NaTx) a hierarchy exists in which specific members (e.g., the *Nv1* cluster) are the major modifiers of the venom phenotype and that their evolution is highly dynamic. As such, understanding the genomic architecture for the expansions and contractions of the *Nv1* cluster among *N. vectensis* populations is critical to identify the mechanisms that underly the evolution of a dominant toxin family and its role in driving variations in the venom phenotype within species.

### Genomic arrangement of *Nv1* loci

To further explore the sequence diversity of *Nv1* among populations of *N. vectensis*, we performed amplicon sequencing of 156 *N. vectensis* individuals from five locations (North Carolina, Massachusetts, New Hampshire, Maine, and Nova Scotia; Fig. 5A). This analysis revealed the presence of 30 distinct *Nv1* variants. Of these 30 distinct *Nv1* variants,

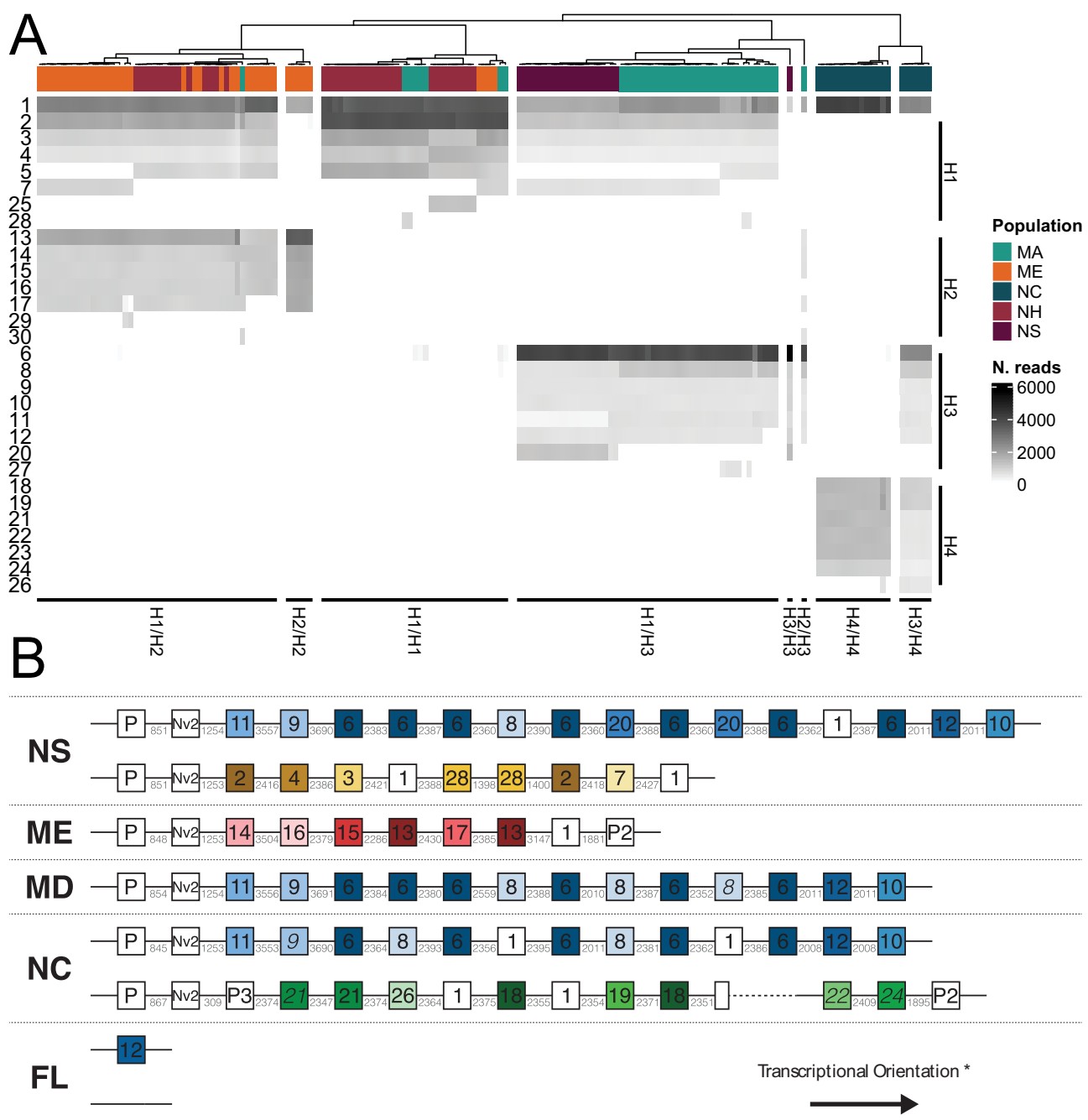

**Fig. 5 | Genomic arrangement of *Nv1* loci. A** Heatmap of *Nv1* paralog abundance in *N. vectensis* samples. Samples are clustered along the *x* axis according to *Nv1* composition and labeled according to the sampled population. *Nv1* paralogs are grouped on the *y* axis according to the inferred haplotype (H1–4). **B** Genomic arrangement of the *Nv1* locus across individuals Each box denotes an *Nv1* paralog with the internal label corresponding to the numeric identifier of the associated amplicon variant. Paralogs are colored according to the core haplotypes identified in the amplicon analyses (Yellow = H1, Red = H2, Blue = H3, Green = H4, White = shared/undetermined) and corresponding multiple sequence alignment and amino acid variants are shown in Supplementary Figs. 3 and 4. Italicized variant labels correspond to the amplicon variant with the lowest pairwise distance to the genomic variant. Pseudogenes (P1, P2, and P3) and *Nv2* copies are labeled separately. Intergenic distances are shown in gray and dashed line indicates a gap of unknown distance. *All loci are shown in the same transcriptional orientation and all variants within each locus share the same orientation.

27 were also found in the transcriptomes of *N. vectensis* from different populations. At the coding sequence level 11 variants are found broadly across multiple different populations, 12 are restricted to the Northeast, and four are found in both North Carolina and New Jersey. Hierarchical clustering of these amplicon-derived *Nv1* paralogs at the DNA level revealed groups of samples that share *Nv1* locus genotypes, from which four "core" haplotypes can be deduced from homozygous individuals (Fig. 5B). While *Nv1.var1* is shared across all haplotypes and

confirmed to be present in all transcriptomes (except for Florida), the remainder of the paralogs are exclusive to a single haplotype. The core haplotypes shared multiple haplotype-specific paralogs although there is evidence of some variability within these haplotypes. For example, heterozygous individuals with an H1 haplotype possess either *Nv1.var5* or *Nv1.var7*.

The distribution of the core haplotypes varies across the range of *N. vectensis*. While H1 is present in 87–100% of individuals from Nova

Scotia, Maine, New Hampshire, and Massachusetts, it is absent from the North Carolina samples (Fig. 5A and Supplementary Fig. 5). Conversely, the H4 haplotype is only present in all samples from North Carolina but absent from all other populations.

Given the significance of Nv1 cluster to the venom phenotype in N. vectensis and its dynamic evolution among populations, we performed long-read sequencing and assembly of Nv1 locus haplotypes for four individuals from different populations (Florida, North Carolina, Maine, and Nova Scotia). Our analysis yielded seven distinct haplotypes, in addition to the haplotype of the recently assembled N. vectensis reference genome[56]. Of the seven haplotypes assembled in this study, six were assembled completely and spanned by single reads (Supplementary Data 18). The number of Nv1 copies per haplotype (including pseudogenes) ranged from 0 to 17 copies. These copy number estimates are higher than the amplification-based analyses due to the presence of mutations in the primer binding sites of some variants (e.g., Nv1.var28). The Floridian haplotype without a single Nv1 copy results from a 30 kb deletion relative to the single-copy haplotype (Supplementary Fig. 6). With the exception of the Florida haplotypes, all haplotypes share a pseudogene and a copy of one paralog (Nv2). The Maine and North Carolina H4 haplotypes also share a pseudogene at the opposite end of the locus.

The composition of the assembled haplotypes corroborates the inferred core haplotypes identified by the amplicon analyses. Of the eight haplotypes (including the ref. [56]) four belong to the H3 haplotype, yet, show extensive variation in the organization of Nv1 paralogs. Considering the arrangement of paralogs and the associated intergenic spacing, expansion of Nova Scotia H3 (blue; Fig. 5B) occurred through serial duplication of single Nv1.var6 copies, and of paired Nv1.var6-Nv1var.20 copies (Nv1.var1 and Nv1.var20 differ by 1 bp intronic indel/mutation). In contrast, North Carolina H3 (blue; Fig. 5B) appears to have undergone a duplication of a Nv1.var8-Nv1.var6-Nv1.var1-Nv1.var6 quadruplet.

There is evidence that transposable elements (TEs) have impacted the Nv1 locus as there is a large insertion into the locus in FL262 with Mutator-like elements (MULEs) at either end (positions 60,895 and 76,278), suggestive of a pack-MULE (Supplementary Data 19). The insertion is found in multiple genomic loci and is ~15 kb in length, which is at the extreme end of pack-MULE size distribution seen in plants[67]. Nevertheless, while the TEs can explain the insertion, they do not appear to explain the deletion of the rest of the Nv1 locus. One plausible explanation for the deletions associated with the Florida haplotypes is the presence of non-b DNA structures. The deletion breakpoints for zero-copy haplotype occurs within 356 bp from a breakpoint associated with the truncated Nv1 found in the Maine and North Carolina haplotypes, suggesting this region may be predisposed to deletions (Supplementary Fig. 7). The breakpoints in this region occur at or adjacent to features known to cause non-canonical (non-b) DNA structures: inverted repeats and poly(G). These structures induce genomic instability and have been associated with large deletions in humans and yeast[68].

## Discussion

Our findings provide a striking example of how gene duplication can impact both micro and macroevolutionary patterns in shifting the venom expression phenotype within and between species of sea anemones. In contrast to the signature of evolutionary constraint acting on toxin genes at the sequence level[37,42,50], here we demonstrate that toxin gene expression evolves rapidly and dynamically, suggesting strong selective forces are acting on toxin gene expression. Phylogenomic analysis supports that gene duplication is likely the underling mechanism that accounts for these adaptive shifts in gene expression. We find similar patterns at the populations scale and show that the dominant toxin in the model sea anemone, N. vectensis, exhibits extreme copy number variation between populations and even individual chromosomes.

Here, we investigated venom evolution in sea anemones across both ecological and evolutionary timescales. Broadly, our results reveal that the expression of a single toxin family dominates the venom expression phenotype of sea anemones and that they can shift between, and even within a species in a highly dynamic manner. Modeling the expression of toxin families confirms that their evolution is best explained by rapid pulses of evolution. Convergent shifts in the dominant toxin are observed to occur, with the venom expression phenotype of species found across superfamilies clustering together. Convergent evolution is often a hallmark of adaptive evolution, thus indicating that the dominant toxin is having an adaptive role required for ecological specialization.

Our findings also show a similar mechanism to venomous snakes[28,53]. In work by Barua & Mikheyev[28], the snake venom phenotype is largely dictated by a single dominant toxin, which explains its low dimensionality and lack of phylogenetic constraint acting on the venom combinations. By comparing our result with those found in snakes we see that selection driving toxin families to become dominant, rather than intrinsic constraints, likely plays the major role in shaping the venom phenotype for both sea anemones and snakes. A similar pattern was reported in cone snails, in which a single toxin superfamily often accounts for >50% of the total conotoxin expression[69]. These dominant toxin superfamilies convergently evolve in a highly dynamic manner, where closely-related species have different dominant toxins[69]. From these results, we suggest that given venom is a polygenic trait in many other venomous animals, a single dominant toxin family is the major dictator of the venom phenotype and the shift in the dominant toxin is likely driven by selection to meet the ecological requirements of these animals.

Our findings provide evidence that a single toxin family dictates the phenotype of venom in sea anemones, while other venom components likely have a more indirect effect. A potential constraint of this phylotranscriptomic approach is the assumption that toxin transcript abundances accurately represent the venom phenotype. The correlation between transcript abundances and protein/peptide expression has been the subject of debate within the venom field[70–73], and more widely (e.g., refs. [74,75]). Nevertheless, we consider our transcriptomic approach robust for the following three reasons: First, a previous quantitative interspecies study did not find evidence of protein-level buffering in venoms that could complicate interspecific comparisons[73]. Second, to avoid known issues with false positives in transcriptomic analyses, we applied stringent filters to restrict our analyses to bona fide sea anemone toxins. Lastly, our work with N. vectensis (ref. [43]; this study) has shown strong congruence between toxin transcript and protein/peptide abundance.

From the transcriptomic observations, we see similarities with the omnigenic model which is a framework to understand the polygenic architecture of complex traits by categorizing groups of complex trait genes as either core or peripheral genes. Proposed by Boyle et al.[1], the value of a given trait is largely determined by the expression level of a few core genes in the relevant tissue, while genes co-expressed likely have a more indirect effect on the phenotype. We see a correlation between the omnigenic model and the venom expression phenotype in which a single dominant toxin family act as core genes that directly affect the venom expression phenotype. Other toxin genes, however, act more like peripheral genes, affecting the venom expression phenotype in a more indirect manner and could possibly be acting synergistically with the dominant toxin. This has previously been shown in various spitting snakes where phospholipase A2 (PLA2) potentiates the dominant toxin, cytotoxic three-finger toxins which accounts for the majority of the protein abundance in their venom profile[76]. Therefore, while all toxin components of the venom may contribute to the heritable variance of the complex trait, the core genes are the major

modifiers of the venom phenotype. It should be noted, however, that the omnigenic model does not perfectly fit venom as a complex trait. Venom is a relatively unique trait in the sense that the toxic cocktail used by the organism is a terminal component of venom production and that toxins are unlikely to impact this complex pathway through feedback loops, while the omnigenic model was conceptualized to understand how networks of genes impact a complex trait. To apply this to the venom phenotype, it would require exploration into the network involved in venom production. To unravel this in sea anemones, comparative transcriptomics of nematocytes and gland cells would be needed. However, applying the omnigenic model to understand the phenotype of the venom cocktail itself still gives us insights into understanding the complex trait by categorizing different toxin families into groups such as core and peripheral genes.

Recent works are unraveling the impact of gene expression on the fitness of an organism. For example, variation at the nucleotide level driving changes in gene expression was shown to be the major modifier of the fitness landscape of protein-coding genes in an experimental setup using the model yeast *Saccharomyces cerevisiae*[77]. These findings indicate that there is greater constraint acting on the sequence of highly expressed genes and highlights that gene expression levels and sequence evolution are interrelated[77]. A recent ground-breaking study by Monroe et al.[78] provides insight into the mechanisms responsible for the evidence that highly expressed genes are under pronounced signatures of constraint. The authors find that differences in genes that are essential have a reduction in the mutation rate by 37% in *Arabidopsis thaliana* and that this reduced mutation rate for essential genes is associated with epigenomic features, such as H3K4me1[78]. In the context of venom evolution, we suspect that the distinction between a toxin family being categorized as either a core gene or peripheral gene may have important implications in the selection pressures acting at the sequence level. This is supported by previous work in different populations of eastern diamondback rattlesnakes (*Crotalus adamanteus*) that revealed that toxin gene expression dynamics, not positive selection at the nucleotide level, was the mechanism for these animals to overcome the resistance of population-specific prey, highlighting the ecological impact and selective pressure acting on toxin gene expression levels[79]. Dominant toxins have been proposed to be essential for broad ecological functions (such as general prey capture), while the peripheral toxins may have more prey-specific functions and are characterized by having greater divergence both at the expression and amino acid sequence levels[80–82].

Taken together with our findings, we report that the dominant toxin dictates the venom phenotype of sea anemones and hypothesize that this phenomenon might be shared across sea anemones, snakes and cone snails as well as other venomous groups, suggesting this is a trend that has evolved convergently among distantly related lineages. We argue that gene duplication is the mechanism that underlies this process.

Gene duplication represents an important mechanism for generating phenotypic variation over ecological and evolutionary timescales through the alteration of gene expression and diversification of variants[83–85]. We find that gene duplication plays a role in shaping the venom phenotype in sea anemones across both micro and macroevolution. The maintenance of clusters of duplicated genes is hypothesized to occur due to conserved regulation of expression. For toxin genes, highly duplicated toxins retained in a cluster could result in increased production of toxin protein due to the transcription of many copies of highly similar or identical genes[12]. In the case of Nv1, this is well-supported by measurements at the transcriptomic and proteomic levels in our current study and previously published works[43]. However, the transcription of *Nv1* varies significantly during the life cycle[43] and in response to a variety of environmental variation such as temperature and salinity[86] and light periodicity[87]. Environmentally elicited expression of *Nv1* differs based on the geographic origin and this

transcriptional variation correlates with CNV, suggesting that gene dosage is the potential mechanism for local adaptation[86] (Supplementary Fig. 8). These results are consistent with snake myotoxins where it has been proposed that selection acts to increase expression as opposed to providing diversity through the permanent heterozygote or multiallelic diversifying selection models[79]. However, these myotoxin analyses excluded sequence variation in the exon responsible for the signal peptide as it is cleaved from the mature toxin. While we also observed low diversity of the mature toxin, consistent with ref. [13], our analyses across multiple populations show non-synonymous variation in the signal and propart peptide sequences. While the functional role of sequence variation in these regions in venom genes has not yet been explored, the amino acid composition and arrangement in signal and propart peptides has been shown to alter translocation, translation and cleavage efficiency[88]. As such, variation in this region of the gene could presumably alter the post-translational regulation of *Nv1*.

The Florida haplotypes raise important questions regarding their origin and the ecology of these populations. The presence of a haplotype without the *Nv1* locus suggests that *Nv1*-less homozygotes may be present in wild *N. vectensis* populations. The *Nv1*-less haplotype could reflect a phenomenon similar to the A-B dichotomy observed in snakes, where two distinct types of venoms exist in a largely mutually exclusive manner[89]. Under this scenario, Florida individuals may have compensated for low *Nv1* copies through the expansion of other toxin genes. However, we find no evidence of compensatory gene family expansion in 11 other known *N. vectensis* toxin genes (Supplementary Fig. 9). Alternatively, the low copy numbers associated with the Florida individual could result from the fitness costs associated with high gene expression. Venom production has a significant metabolic cost in *N. vectensis*[86] and *Nv1* is expressed by almost two orders of magnitude higher compared to the other toxins[43]. Thus, reduced venom capacity in a population at the upper thermal limit of the species range could potentially reflect the metabolic strain of venom production. However, summer temperatures in the South Carolina habitat are relatively similar to the ones in the Florida habitat. Instead, we suggest that such a massive reduction in toxin production as observed here should be associated with at least some differences in prey and/or predator composition and abundance between the Florida and South Carolina habitats as loss of defense or ability to predate with venom can be highly deleterious.

The exclusivity of paralogs to particular haplotypes suggests that recombination between contemporary haplotypes does not occur or is rare enough that it is beyond our limits of detection with these samples. The observed lack of recombination does not appear to result from the absence of heterozygous individuals as they are present in all populations, although, it is important to note that recombination between contemporary haplotypes could occur but its prevalence may be impacted by other factors such as selection. Nevertheless, this lack of evidence for recombination between core haplotypes helps provide insight into the mechanisms governing expansion and contraction within haplotypes. We observe substantial variation in the copy number, composition, and organization of paralogs within haplotypes including tandem duplications of singlet, duplet, and quadruplet *Nv1* paralogs (Supplementary Fig. 10). Furthermore, the expansion of different paralogs indicates that multiple independent expansion events have likely occurred at the same locus. The expansion and contraction of *Nv1* paralogs within core haplotypes in *N. vectensis* could be driven by non-allelic homologous recombination (NAHR) or replication slippage. NAHR is commonly associated with CNVs, including toxin genes, and within a single *Nv1* haplotype, there is a sufficient substrate for NAHR with regions of high sequence homology extending over >300 bp. In snakes, transposable elements have been proposed as the NAHR substrate[90,91]; however, this does not appear to be the case for the *Nv1* locus as TEs are largely absent from within the *Nv1* locus. If

NAHR is the mechanism driving the expansion and contraction of the *Nv1* haplotypes, it is unclear why it would not occur across haplotypes because sufficient NAHR substrate is evident between haplotypes despite the presence of haplotype-specific paralogs. An alternative hypothesis would be that expansions and contractions at the locus are a result of backward replication slippage[92]. We suggest that this mechanism is more likely responsible for the tandem duplications at the *Nv1* locus as there is the sufficient substrate, the duplication sizes are consistent with past observations of replication slippage, and importantly, would maintain the strong haplotype structure.

The presence of *Nv1.var12* in the single-copy Florida haplotype indicates that it is most closely related to the core haplotype H3. However, considering that this copy is not at the end of the locus, it suggests that even the single-copy Florida haplotype is at least two mutational steps from its closest relative and warrants further exploration for intermediate haplotypes in populations in the southeastern United States (e.g., Georgia). Analysis of the genomic context of the *Nv1* locus in the Florida haplotypes suggests that NAHR is unlikely to be the cause of these extreme contractions (Supplementary Fig. 6) and indicates that other processes are involved in the evolution of the *Nv1* locus.

The homogeneity of *Nv1* genes in the gene cluster has previously been hypothesized to maintain sequence similarity of duplicated genes through concerted evolution[13]. Later analyses of *Nv1*-like paralogs that translocated out of the cluster, which accrued proportionally more sequence divergence, further supported a hypothesis for concerted evolution of the *Nv1* cluster[66]. Toxin genes in other cnidarians also showed patterns of highly similar genes resulting from lineage-specific duplication events[13,38], suggesting concerted evolution may be common in the expansion of toxin families. Here, evidence for concerted evolution at the *Nv1* locus is confounded by our analysis of the spatial organization of *Nv1* genes in the cluster. First, although the reference haplotypes for the current and past genome assemblies contain a numerically overrepresented sequence, this is not a feature of all of the *Nv1* haplotypes. Second, the tandem arrangement of groups of paralogs (doublets, quadruplets) with consistent intergenic spacing might suggest that some of the similarities in loci is due to more recent duplications that retain the evolutionary history of the ancestral loci prior to duplications rather than homogenization of the array.

An alternative or additional hypothesis to concerted evolution for this locus is the birth–death model that has been proposed for other venom genes including *Nv1* paralogs that have escaped the *Nv1* locus[66]. Here, new gene copies arise through repeated duplications with some copies retained in the genome, while others become non-functional through mutation or are deleted[93]. Our analyses of the composition and spatial organization of *Nv1* genes demonstrate repeated duplications of genes and pairs of genes, providing support for the birth process. Furthermore, we also observed pseudogenes highlighting that not all genes are retained after duplication. Nevertheless, it is important to note that their number is very small compared to the seemingly functional copies. Due to the absence of an ancestral sequence, it is not possible to conclusively determine the extent of gene losses versus gene gains, however, the single-copy and *Nv1*-less haplotypes in Florida could represent a rapid gene death process. This would be consistent with other venom gene families where large deletions of genes have been observed[91].

Overall, our observations across macro- and microevolutionary timescales demonstrate that a single toxin family dictates the complex venom phenotype among sea anemones. Gene duplication underlies which toxin family becomes dominant through a process of increasing gene expression and this process is highly dynamic resulting in the rapid evolution of the venom phenotype across different species. High gene turnover rates of the dominant toxin family are found even within species, further signaling that strong selective forces are acting on toxin gene expression. Finally, as we see a similar trend is found in other venomous species, we hypothesize that gene duplication-driven dominance by a single toxin family is a fundamental process shaping the venom phenotype.

## Methods

### Phylotranscriptomics

We analyzed transcriptomes from 29 sea anemone species, spanning three of the five Actiniarian superfamilies (Actinioidea, Edwardsioidea, and Metrioidea). These transcriptomes that were sampled from either multiple tissues or tentacles were downloaded from NCBI SRA using FASTQ-DUMP in the SRA toolkit. Raw reads retrieved were assessed for quality and trimmed using Trimmomatic[94]. Trinity was used to assemble transcriptomes de novo from the filtered raw reads[95]. BUSCO (v4) was used to validate the quality and completeness of the transcriptomes[96]. Transcripts corresponding to toxins were identified using previously established methods[37], and then manually curated. Briefly, predicated open-reading frames encoding proteins for transcripts from each transcriptome was identified using ORF-finder (https://www.ncbi.nlm.nih.gov/orffinder/) and BLASTp (E-value 1e-01) performed against the swiss-prot database. Top hits against sea anemones toxins characterized in the Tox-Prot database[97] were retained and used to determine the presence of a signal peptide using SignalP (v5.0[98]). These sequences were then characterized into toxin families and aligned using[99] to retain only those with conserved cysteine frameworks are essential residues. Toxin families used in this analysis included only those that have been functionally characterized as toxins in multiple sea anemone species or shown to be localized to venom producing cells using multiple experimental approaches.

Toxin expression data were generated using software leveraged in the Trinity package (v > 2.2[100]). This included individual reads being mapped back to reference de novo transcriptome assemblies independently for each species using Bowtie2[101], and abundance estimated using RSEM[102]. Normalized abundance estimates of the transcript were calculated and corrected for their length to generate TPM values. Finally, we calculated the cumulative TPM values for each toxin family and the venom phenotype was generated as the percentage that each family contributes.

Transcripts with TPM values greater than zero were retained and their predicated open-reading frame was detected using ORFfinder (https://www.ncbi.nlm.nih.gov/orffinder/). Open-reading frames encoding proteins >100 amino acids in length were retained and redundant sequences with >88% similarity were removed to produce a predicted proteome for the 29 transcriptomes using CD-HIT[103].

Single-copy orthologs were identified using DIAMOND within the Orthofinder package[104]. This identified 138 single-copy orthologs that were individually aligned using MAFFT[99] and nucleotide alignment was generated using Pal2Nal using the coding sequence[105]. Aligned orthologs were concatenated and imported into IQ-TREE to determine the best-fit model of evolution[106]. The JTT model with gamma rate heterogeneity, invariable sites, and empirical codon frequencies were selected, and a maximum-likelihood tree was generated using 1000 ultrafast bootstrap iterations. An ultrametric tree was generated using by calibrating the maximum-likelihood tree Chronos function within the R package Ape using minimum and maximum age of root set to 424 and 608 million years ago[107,108]. Different calibration models were tested, including correlated, discrete, and relaxed models, with the discrete model determined to be the best fit.

### Phylogenetic covariance analysis

PCA was performed as per ref. [28] using the R package MCMCglmm[109] with a multivariate model being used and toxin families as the

response variable. 20 million iterations were used, which included burnin and thinning values of 1 million and 1500, respectively. The phylogenetic signal was determined as previously described[28,110]. Principal component analysis was used by obtaining the phylogenetic covariances generated from the MCMCglmm analysis. Given sea anemones have a decentralized venom system, we tested whether the tissue type used to generate the raw reads significantly impacted the phylogenetic effect[25,26].

### Modeling the ancestral states and modes of evolution acting on sea anemone venom

The R packages SURFACE and pulsR were used to test the models of evolution[111]. Evidence of phenotypic convergence was tested using SURFACE. The pulsR package was used to test the evolution of venom expression phenotype as either through a model incremental evolution or through pulsed evolution as modeled using the Lévy process[54]. The ancestral venom expression phenotype was reconstructed using *fastAnc* in the Phytools package[112].

### Macro and microsynteny

Homologous chromosomes were found among the three genomes to determine macrosynteny. This was achieved by identifying 3767 single-copy orthologs using proteins annotated from all three genomes. The genomes of *N. vectensis*, *S. callimorphus* and *A. equina* were all investigated for the presence of NaTx and KTx3 toxins. Toxins from these genomes were identified using transcripts previously assembled using Trinity and mapped to the genome using Splign online software (https://www.ncbi.nlm.nih.gov/sutils/splign/splign.cgi). The chromosomal locations for the single-copy orthologs were compared to generate a broad macrosyntenic map of chromosomes among sea anemone genomes. The microsynteny neighboring NaTx/KTx3 loci was investigated by using BLAT from 30Kb upstream and downstream of the loci as well as comparing the 3 protein-coding genes upstream and downstream from NaTx/KTx3 loci. The NaTx/KTx3 copies identified from the genomes of the three species were clustered with publicly available copies previously used for evolutionary analyses[50], using CLANS software[113] with default settings and 300,000 rounds.

### Population transcriptomics

**Animal collection.** Adult *N. vectensis* were collected from estuaries along the Atlantic coast of the United States and Canada. We collected 20 individuals from five locations (Crescent Beach, Nova Scotia; Saco, Maine; Wallis Sand, New Hampshire; Sippewissett, Massachusetts; Ft. Fisher, North Carolina) in March 2016, and an additional 10 individuals/month from three of these locations (Saco, Maine; Wallis Sands, New Hampshire; Sippewissett, Massachusetts) in June and September 2016. Individuals were stored in RNAlater and stored at −20 °C prior to nucleic acid extraction for qPCR and amplicon analyses. At each collection, additional individuals were transported to UNC Charlotte and cultured in the laboratory under standard laboratory conditions (15 parts per thousand artificial seawater, room temperature, fed freshly hatched Artemia 2-3 times per week). In addition, eight adult *N. vectensis* collected near St. Augustine, Florida were kindly provided by Lukas Schäre (University of Basel). From these laboratory populations, we selected four individual anemones to grow clonal lines for long-read sequencing; single individuals from Nova Scotia, Maine, North Carolina, and Florida were grown and bisected to generate the lines.

To investigate the population-level comparison of venom among *N. vectensis*, transcriptomics was performed. Multiple individuals from nine locations in North America were collected. This included the same locations as mentioned above in Florida, Massachusetts, Maine, North Carolina, New Hampshire, Nova Scotia and as well as New Jersey (Brigantine), Maryland (Rhode River), and South Carolina (Georgetown). Individuals from these locations were brought back to the lab and allowed to acclimatize for 2 weeks.

Total RNA was extracted from pools of three whole specimens per site from nine different locations using RNeasy Mini Kit (Qiagen, USA). Quality and integrity of extracted RNA was assessed using Bioanalyzer 2100 (Agilent, USA) using an RNA nano chip (RIN > 8). Sequencing libraries were prepared using the Kapa Stranded mRNA-seq kit (Roche, Switzerland) and sequenced on an Illumina HiSeq 4000 using 150 bp paired-end chemistry performed at Duke Center for Genomic and Computational Biology (Durham, NC, USA). Raw reads from each population were cleaned using Trimmomatic[94] to retain only high-quality reads and to remove non-biological sequence and assembled into nine transcriptomes using the Trinity 2.6.6[95]. To assess the completeness of de novo transcriptomes, BUSCO (v3.0) was performed on each transcriptome to assess the completeness of each assembly, by determining the percentage of full-length sequences in each transcriptome corresponding to a conserved set of metazoan orthologs[114].

Comparative transcriptomics were performed to reconstruct the phylogenetic relatedness of *N. vectensis* populations across North America. For each transcriptome, open-reading frames were identified using ORFfinder and translated using Transeq. Redundant sequences with >88% sequence similarity were removed using CD-HIT[103]. Protein sequences >100 amino acids in length were used to identify single-copy orthologs using OrthoFinder[115] and leveraged using DIAMOND[116]. In addition, we added *S. callimorphus* and *Edwardsiella carnea* as outgroups to the *N. vectensis* populations. This resulted in 2589 single-copy orthologs shared among the 11 transcriptomes. Protein sequences for each single-copy ortholog were individually aligned using MAFFT[99]. Protein alignments were then concatenated and imported into IQ-TREE, and the best-fit model of evolution selected using ModelFinder[106], and posterior mean site frequency models were used to reduce long-branch attraction artefacts[117]. A maximum-likelihood phylogenetic tree was generated using 1000 ultrafast bootstrap iterations.

Variations of toxins from the NaTx gene family in *N. vectensis* (*Nv1*, *Nv2*, *Nv3*, *Nv4*, *Nv5*, *Nv6*, and *Nv7*) were investigated among the different populations using multiple approaches. Initially, BLASTp was performed to identify toxins using ORFs from the transcriptomes against a custom database consisting of all known sequences from the *Nv1* gene family. Sequences with a significant hit (*E*-value 1e-05) were then manually curated to determine the presence of a signal peptide and conserved cysteine framework. In addition, *Nv1* copies have been previously reported to be massively duplicated (with at least ten copies previously reported) and highly homogenous in the genome of *N. vectensis*[13]. For these reasons, additional approaches were required to capture these limited variations of *Nv1* copies among the populations. To achieve this, cleaned raw reads were mapped using Bowtie2 plugin in Trinity using default settings[95,101] to the *N. vectensis* gene models with *Nv1* reduced to a single copy[86]. Paired-end reads mapping to *Nv1* were then extracted and aligned to the *Nv1* gene model using MAFFT[99], and a new consensus *Nv1* sequence generated for each mapped paired-end read using cons in EMBOSS. Identical *Nv1* sequences were then clustered using CD-HIT-EST[103] and only the top most abundant sequences that accounted for 70% of the total number of sequences or had a minimum of 10 identical copies were retained. Florida sample was an exception in which only the most abundant sequence was retained as it had four identical copies. The open-reading frame was identified and redundant coding sequences with removed to give a representation of allelic variation in *Nv1* in different populations. To obtain a allelic variation of *Nv3*, mapped paired-end reads that had a *Nv3* signature (AAACGCGGCTTTGCT, which encodes for KRGFA, as opposed to *Nv1* AAACGCGGCATTCCT which encodes for KRGIP) were extracted and aligned to *Nv3* coding sequence using MAFFT[99]. The most abundant consensus sequences that accounted for 70% of the total number of sequences or had a minimum of 10 identical copies were retained, and redundant coding sequences removed.

## Population proteomics

Semi-quantitative MS/MS analysis was performed using adults (four replicates, each made of three individuals) from both North Carolina and Florida. Samples were snap frozen and lysed using in 8 M urea and 400 mM ammonium bicarbonate solution. Lysed samples were centrifuged (22,000 × $g$, 20 min, 4 °C) and supernatant collected. Protein concentrations were measured with BCA Protein Assay Kit (Thermo Fisher Scientific).

**Sample preparation for MS analysis.** Ten micrograms of protein were dissolved in 100 µl of 8 M urea, 10 mM DTT, 25 mM Tris-HCl pH 8.0 for 30 min at 22 °C. Iodoacetamide (55 mM) was added and followed by incubation for 30 min (22 °C, in the dark). The samples were diluted with 8 volumes of 25 mM Tris-HCl pH 8.0 followed by the addition of sequencing-grade modified Trypsin (Promega Corp., Madison, WI) (0.4 µg/ sample) and incubation overnight at 37 °C. The peptides were acidified by the addition of 0.4% formic acid and transferred to C18 home-made stage tips for desalting. The peptide concentration was determined by absorbance at 280 nm and 0.3 µg of peptides were injected into the mass spectrometer.

**nanoLC-MS/MS analysis.** nanoLC-MS/MS analysis was performed as previously described in ref. [118] with the exception that peptides dissolved in 0.1% formic acid were separated without a trap column over an 80 min acetonitrile gradient run at a flow rate of 0.3 µl/min on a reverse phase 25-cm-long C18 column (75 µm ID, 2 µm, 100 Å, Thermo PepMapRSLC). The instrument settings were as described in ref. [119].

**MS data analysis.** Mass spectra data were processed using the MaxQuant computational platform, version 2.0.3.0. Peak lists were searched against an NVE FASTA sequence database (https://figshare.com/articles/Nematostella_vectensis_transcriptome_and_gene_models_v2_0/807696). The search included cysteine carbamidomethylation as a fixed modification, N-terminal acetylation, and oxidation of methionine as variable modifications and allowed up to two miscleavages. The "match-between-runs" option was used. Peptides with a length of at least seven amino acids were considered and the required FDR was set to 1% at the peptide and protein level. Relative protein quantification in MaxQuant was performed using the LFQ algorithm[120]. MaxLFQ allows accurate proteome-wide label-free quantification by delayed normalization and maximal peptide ratio extraction.

Statistical analysis ($n = 4$) was performed using the Perseus statistical package, Version 1.6.2.2[121]. Only those proteins for which at least three valid LFQ values were obtained in at least one sample group were accepted and log2 transformed. Statistical analysis by Student's $t$ test and permutation-based FDR ($P$ value <0.05). After the application of this filter, a random value was substituted for proteins for which LFQ could not be determined ("Imputation" function of Perseus). The imputed values were in the range of 10% of the median value of all the proteins in the sample and allowed the calculation of $P$ values. To test if proteomes were comparable, we performed linear regression between the Florida and North Carolina samples. Proteins with non-zero LFQ values in at least one sample for each population were used and transformed per million.

## Population genomics

**Quantification of Nv1 copy number.** We used quantitative PCR to determine the number of *Nv1* copies in individuals collected from each location using hydrolysis probe-based quantitative PCR. DNA for anemones from each location was isolated with the AllPrep DNA/RNA kit (Qiagen) using the manufacturer's protocol. Primers and hydrolysis probes were designed for *Nv1* and *Catalase* using Primer3[122]. The hydrolysis probes contained distinct fluorophores for each gene in addition to 3' and internal quenchers (*Nv1* = 5' Cy5/

TAO/3' IBRQ; Cat = 5' 6-FAM/ZEN/3' IBFQ). Amplifications were performed in an Applied Biosystems 7500 Fast Real-Time PCR System using the Luna Universal Probe Mix (NEB). We evaluated the performance of the qPCR primers and probes using a DNA concentration gradient spanning 0.1–100.0 ng/reaction in single gene reactions as well as in a multiplex reaction. The efficiencies of both gene assays were within the recommended range (90–110%), comparable across the concentration gradient, and consistent between the single and multiplex reactions. As such, we amplified *Catalase* and *Nv1* in triplicate multiplex reactions for each sample, and each 96-well plate contained samples from all populations. In addition, each plate contained triplicate reactions of a reference sample of known copy number from Florida (diploid copy number = 1; derived from genome assembly), a no template control (NTC), and two samples to monitor variability between plates. The diploid copy number was estimated using the ΔΔCt approach with Catalase as the single-copy control gene and the Floridian sample of known copy number as our reference sample. There was no amplification observed in any NTCs.

The Cq values were determined automatically in the Applied Biosystems software. We filtered individuals from the qPCR results where the Cq values for either gene was outside of the range used for efficiency estimation (two individuals), and filtered individual reactions where the Cq values deviated by more than 0.2 Cq between any of the triplicate reactions for either gene (6/549 reactions). As we used multiplex reactions, mean ΔCt was calculated as the mean of ΔCt across individual reactions. We performed a two-way ANOVA (diploid copy number ~ population * plate) in cab package R using Type II SS (to account for the unbalanced design) to test for the effect of population on diploid *Nv1* copy number while accounting for any potential batch effects. The ANOVA tests revealed no significant effect of plate or population-by-plate on our copy number estimates. Tukey HSD post hoc tests (alpha = 0.05) were performed using the agricolae package[123].

**Amplification and sequencing of Nv1.** We designed primers to amplify *Nv1* loci from genomic DNA for sequence analysis with the Illumina MiSeq. Primers were designed to amplify the full coding sequence for *Nv1* and minimized mismatches with SNPs identified between known *Nv1* variants. Primers contained the adapter overhang for Nextera Indexing. PCRs were performed with HiFi HotStart ReadyMix (Kapa Biosciences) using the following conditions: 95 °C−3 min; 8 x (95 °C−30s, 55 °C−30 s, 72 °C−30 s), 72 °C−5 min. PCR products were purified with Ampure XP beads (Beckman Colter). Successful amplification of the anticipated product size was verified by gel electrophoresis. Amplicons from each sample were quantified by Qubit (Thermo Fisher Scientific) for normalization. Equal concentrations of each sample were pooled with 5% PhiX for sequencing using a MiSeq v3 reagent kit (600 cycles). We used mothur v1.44.3[124] to join overlapping reads to make contigs that were subsequently filtered to remove amplicons outside of *Nv1* size expectations (300–500 bp) and with ambiguous bases. Cutadapt v2.6[125] was subsequently used to remove primer sequences. We randomly subsampled the FASTA files to a depth of 14,800 reads, with four samples removed from future analyses due to insufficient reads. In order to distinguish biological sequence variation from methodological artifacts (e.g., PCR and sequencing errors), we identified a list of variants based on a minimum sample read abundance of 100 and presence in more than one individual. A heatmap of relative abundance of variants across samples was generated using the Complex-Heatmap package[126] in R. Hierarchical clustering of samples was performed using Spearman rank correlation as the distance measure.

**PacBio and nanopore sequencing.** High-quality DNA was extracted from individual clone lines from four geographic locations (Nova Scotia, Maine, North Carolina, and Florida) using a previously described extraction protocol[127]. This protocol was adapted for HMW DNA by the addition of tissue grinding in liquid nitrogen, decreasing the incubation time and temperature to one hour at 42 °C, increasing the elution time to 24 h, and the use of wide-bore tips throughout the protocol.

For Nova Scotia and Florida anemones we sequenced DNA from single genotypes with PacBio technology. DNA was shipped to Brigham Young University (Provo, UT, USA) for quality check with pulse-field capillary electrophoresis followed by CLR library construction and sequencing (PacBio Sequel II). The unique molecular yields were 38 Gb and 123 Gb, with the longest subread N50s of 35 kb and 28 kb, respectively. PacBio reads were assembled into contigs using Canu v2.0[128], configured to assemble both haplotypes at each locus separately. Two rounds of polishing were applied to each assembly by aligning raw PacBio data using pbmm2 (v1.3.0) and using the multi-molecule consensus setting of the Arrow algorithm implemented in gcpp (v.1.9.0)[129]. Transposable elements annotations for the PacBio assemblies, in addition to the Maryland reference, were generated by EDTA v1.9.6[130] using a combined fasta file containing all three assemblies.

For Maine and North Carolina anemones, short DNA fragments were removed using the short read eliminator kit (Circulomics). Libraries were prepared for Nanopore sequencing using the ligation sequencing kit (LSK109) and sequenced on a single MinION flow cell (R9.4.1; Oxford Nanopore Technologies). The Nanopore long reads were basecalled using guppy (v4.5.2), assembled into contigs using Canu v2.1[128], and *Nv1* contigs polished using Racon v1.4.21[131]. For the Maine sample, only one *Nv1* haplotype was assembled. Evaluation of the intergenic spacing of *Nv1* copies in raw reads based on BLASTn searches was consistent across all reads suggestive of a homozygous individual. In contrast, evaluation of the North Carolina raw reads showed reads could be split into two separate groups based on disparate intergenic spacings. These two sets of reads were assembled separately.

For this study, we have only focused our analysis on the contigs corresponding to the *Nv1* cluster. These contigs and their respective *Nv1* copy number and localization were identified using BLASTn searches against the assemblies. Pseudogenes were identified as copies as *Nv1* copies with premature stop codons and truncated mature peptide sequences. An analysis of the remaining portions of the genome for each clone line will be reported in a future publication.

Expression of *Nv1* for individuals originally collected from Florida was quantified with nCounter technology. This approach was identical to methods reported for quantification of *Nv1* for *N. vectensis* from other geographic locations reported in Sachova et al. [86]. Briefly, individuals were acclimatized at 20 °C for 24 h in the dark in 15‰ artificial seawater (ASW). Individuals were subsequently exposed to one of three temperatures in the dark: 20 °C (control), 28 °C, and 36 °C for 24 h. Animals were placed into tubes and frozen to obtain three replicates for each condition, two animals/replicate. Extracted RNA was shipped for analysis using the nCounter platform (NanoString Technologies, USA; performed by MOgene, USA) for expression of *Nv1* using the same custom probe previously reported.

## Reporting summary
Further information on research design is available in the Nature Portfolio Reporting Summary linked to this article.

## Data availability
Raw sequencing data for *N. vectensis* populations have been submitted to the NCBI SRA database for transcriptomics (BioProject: PRJNA831625), amplicon sequencing (BioProject: PRJNA836916) and genomics (BioProject: PRJNA844989). Proteomics from North Carolina and Florida populations has been submitted to the proteome exchange (PXD034383). Sequences used in this study have also been uploaded as FASTA files to figshare (https://doi.org/10.6084/m9.figshare.20115719.v1). Accession numbers for data used in this project for the phylotranscriptomics can be found in Supplementary Data 1. Source data are provided with this paper.

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

## Acknowledgements

The authors are grateful to Dr. William Breuer (Mass Spectrometry Core Facility, The Hebrew University of Jerusalem) for his help with proteomics and Dr. Peter Prentis (Queensland University of Technology) for providing sea anemone photos. This work was supported by the Lady Davis fellowship to J.M.S., National Science Foundation fellowship 1924498 to E.G.S., Israel Science Foundation grant 636/21 to Y.M., incentive funding from the CIPHER Center at UNC Charlotte to A.M.R., and Binational Science Foundation program with the National Science Foundation grants 1536530 and 2020669 to Y.M. and A.M.R.

## Author contributions

Conceptualization: E.G.S., J.M.S., J.M., A.M.R., and Y.M.; computational analysis: E.G.S., J.M.S., J.M., A.S., and G.A.; Experimental analysis: E.G.S., J.M.S., J.M., M.Y.S., and M.L.; writing—original draft: E.G.S., J.M.S., A.M.R., and Y.M.; writing—review and editing all authors; supervision: J.M.S., A.M.R., and Y.M.; funding acquisition: E.G.S., J.M.S., A.M.R., and Y.M.

## Competing interests

The authors declare no competing interests.
