## [Peer Review File · Nature Communications]

Dominant toxin hypothesis: unravelling the venom phenotype across micro and macroevolutionReviewers' Comments:

Reviewer #1:

Remarks to the Author:

This manuscript documents that a particular toxin family's expression and high copy numbers is an important strategy in the evolution of toxins within sea anemones. Using comparative transcriptomics and phylogenetic analysis along with genomic synteny analysis of three sea anemones genomes in publicly available datasets they evaluate broader patterns of toxin expression across sea anemones. The authors also use a variety of population-level analyses of Nv1 toxins in native *Nematostella* populations. This is an excellent study, as they have shown that dominance in specific toxin families occurs in a consistently and phylogenetically-independent manner. This work will be highly relevant within the venom field and more broadly to those interested in the evolution of expression dynamics in other systems. My comments are relatively minor.

- There seems to be a slight disconnect in topics from paragraphs three to four. While the first three paragraphs do a nice job mechanistically of describing the role of gene expression dynamics and CNV in the evolution of toxin phenotypes, the fourth paragraph does not really go back and address these processes and their role broadly in sea anemones venom. Notably, previous work that has shown gene duplication is a relevant and importance feature particular in sea anemones venoms is absent. Can a brief addition to include some of these previous studies be included?

- The introduction makes reference to toxins in sea anemones being present in both nematocysts and gland cells, but no specific description of Nv1 being specifically localized to ectodermal gland cells in *Nematostella* is in the text. Given the importance of this system within the study, it seems relevant to note this (even briefly) within the main text.

- Throughout this work the authors make reference to the "venom phenotype" when referring to the RNA-seq expression clustering for different toxin families in each sea anemone species. This becomes slightly confusing when venom phenotype would also broadly be referring to peptide/protein expression and overall venom activity, that is the gene expression patterns codes for phenotype (gene expression is not explicitly the phenotype). In the text, "venom phenotype" seems to be used interchangeably between these two definitions, in particular within the results of the manuscript and within the discussion. Perhaps "venom expression phenotype" would be a clearer descriptor?

- Fourth paragraph, second sentence - "anemone toxin families"?

- Fourth paragraph, third sentence - Extra "and" before potassium channel toxins?

- It may be useful in the first mention of *Nematostella vectensis* to add the complete species name: "*Nematostella vectensis* Stephenson, 1935." This could also be applied to the other species named in this work.

- Supplementary Figure 6: Can the image quality of this figure be improved? It seems to be relatively blurry compared to the other figures.

Reviewer #2:

Remarks to the Author:

This is an interesting and clearly presented study of venom toxin evolution in sea anemones. It offers important new insights into patterns of venom variation within and between species, and the processes responsible for them. The authors used a battery of complementary techniques to generate these in-depth insights. I enjoyed reading this paper. I should point out that I lack firsthand experience in most of the techniques used in this study, so my technical evaluation should be considered that of an interested consumer, not an expert.

I have only a few comments.

As a general comment, this paper makes its inferences on the basis of venom transcriptomes. It is

known mostly from other taxa (but also sea anemones: Madio et al., 2017, J. Proteomics 166) that transcriptomes are imperfect (sometimes very) predictors of proteomic venom profiles. I think that the authors should highlight this constraint of the study, and if possible if such data exists, outline why they think that these transcriptomic insights are likely to be reliable proxies for actual venom composition.

Page 3, 2nd paragraph: "high mutation rates of duplication"; should this be "high mutation rates of duplicated genes"?

Page 5, 2nd paragraph: "macro-" instead of "macro"; "undergoes" instead of "undergo". I will not identify other minor language errors. I leave it to the authors to make the necessary corrections throughout the manuscript.

Page 7, figure 1: Actinioidea, Edwardsioidea, and Metrioidea are superfamilies, not families. It would also be useful to non-expert readers to get a little background so they can appreciate how broadly these samples represent actiniarian diversity. Also please indicate for fig. 1b that the colours correspond to those of the toxins in fig. 1a.

Reviewer #3:

Remarks to the Author:

What are the noteworthy results?

This well written manuscript highlights an emerging theory in venom research concerning gene expression and regulation of dominant toxin types in the venom arsenal. Here they identify the dominant toxin in sea anemones, Nv1, similar to the dominant snake toxins, three-finger toxins (TFTx), Phospholipase A2 (PLA2) and snake venom metalloprotease (SVMP). Using phylogenomic and transcriptomic tools the research team provides convincing evidence for their claim that "gene duplication-driven dominance by a single toxin family is a fundamental process shaping the venom phenotype."

Will the work be of significance to the field and related fields?

The work is significant to the venom field and follows on the heels of a recent finding in Snakes by Barua & colleagues (2019, 2021) where they use similar techniques to demonstrate that snake toxins are made up a few dominant families and gene regulation and duplication is the mechanism driving venom evolution.

This will have significant implications in related fields of evo-devo, molecular and cellular biology as at the core is the study of how complex traits evolved and the characterization of novelty in gene expression.

How does it compare to the established literature? If the work is not original, please provide relevant references.

The manuscript is well cited and reflects the established literature driving the field.

Does the work support the conclusions and claims, or is additional evidence needed?

The weakest part of the manuscript is discussion on the genomic arrangement of the Nvi loci as part of the role of gene duplication in venom phenotype. This is very speculative and the evidence provided is not conclusive. I recommend shortening this section, specifically the discussion on the Florida haplotypes. This can be significantly condensed.

Are there any flaws in the data analysis, interpretation and conclusions?

The only flaw is stated by the authors as they have considered it. That is, in order to truly prove the omnigenic model for venom evolution as they are proposing they would need to do investigate the gene regulatory network involved in venom production by doing both computational analyses (using comparative transcriptomics of nematocytes and gland cells) followed by in vivo genetic manipulation of the gene regulatory network to confirm the computational findings. Currently there are few in vivo model systems for conducting this work, anemones are among the few, but this is beyond the scope of this publication. It is where the venom field has to go to support these claims about dominant toxin theories. If you knock out the dominant toxin gene, what are the implications for the venom arsenal and it's function and can this be inherited for several generations?

Another minor flaw is that there are no images of the anemones used in the study. Adding representative images of the three families used in Figure 1 would go a long way in terms of science communication and grabbing attention of the reader.

Do these prohibit publication or require revision?

No.

Is the methodology sound? Does the work meet the expected standards in your field?

Yes.

Is there enough detail provided in the methods for the work to be reproduced?

Yes.

Dominant toxin hypothesis: unravelling the venom phenotype across micro and macroevolution - Response to Reviewer #1

Reviewer comment: This manuscript documents that a particular toxin family's expression and high copy numbers is an important strategy in the evolution of toxins within sea anemones. Using comparative transcriptomics and phylogenetic analysis along with genomic synteny analysis of three sea anemones genomes in publicly available datasets they evaluate broader patterns of toxin expression across sea anemones. The authors also use a variety of population-level analyses of Nv1 toxins in native *Nematostella* populations. This is an excellent study, as they have shown that dominance in specific toxin families occurs in a consistently and phylogenetically-independent manner. This work will be highly relevant within the venom field and more broadly to those interested in the evolution of expression dynamics in other systems. My comments are relatively minor.

Response to reviewer: We would like to thank the reviewer for taking the time to review our manuscript. We are happy that they found our manuscript of broad interest and suitable for publication in Nature Communications. We have addressed their minor comments below:

Reviewer comment: There seems to be a slight disconnect in topics from paragraphs three to four. While the first three paragraphs do a nice job mechanistically of describing the role of gene expression dynamics and CNV in the evolution of toxin phenotypes, the fourth paragraph does not really go back and address these processes and their role broadly in sea anemones venom. Notably, previous work that has shown gene duplication is a relevant and importance feature particular in sea anemones venoms is absent. Can a brief addition to include some of these previous studies be included?

Response to reviewer: We have amended the text in paragraph four to highlight past studies on the relevance of gene duplication in sea anemone venom.

“Among cnidarians, sea anemone venom is arguably the most well-characterized (Prentis et al., 2018) and past research has shown that toxin gene duplication is an important feature in these organisms (Moran, Weinberger, Sullivan, et al., 2008; Surm, Smith, et al., 2019; Surm, Stewart, et al., 2019)”

Reviewer comment: The introduction makes reference to toxins in sea anemones being present in both nematocysts and gland cells, but no specific description of Nv1 being specifically localized to ectodermal gland cells in *Nematostella* is in the text. Given the importance of this system within the study, it seems relevant to note this (even briefly) within the main text.

Response to reviewer: We agree with the reviewer that this important information was missing from the previous submission and we have amended the introduction to address this omission.

“Located in the ectodermal gland cells (Moran, Genikhovich, et al., 2012a), this sodium channel toxin is the major component of the N. vectensis venom”

Reviewer comment: Throughout this work the authors make reference to the “venom phenotype” when referring to the RNA-seq expression clustering for different toxin families in each sea anemone species. This becomes slightly confusing when venom phenotype would also broadly be referring to peptide/protein expression and overall venom activity, that is the gene expression patterns codes for phenotype (gene expression is not explicitly the phenotype). In the text, “venom phenotype” seems to be used interchangeably between these two definitions, in particular within the results of the manuscript and within the discussion. Perhaps “venom expression phenotype” would be a clearer descriptor?

Response to reviewer: We thank the reviewer for highlighting this potential source of confusion. We have used their suggestion to clearly distinguish between the peptide/protein and gene expression phenotype.

Reviewer comment: Fourth paragraph, second sentence - “anemone toxin families”?

Response to reviewer: We have amended the text as requested.

Reviewer comment: Fourth paragraph, third sentence - Extra “and” before potassium channel toxins?

Response to reviewer: We have amended the text as requested.

Reviewer comment: It may be useful in the first mention of *Nematostella vectensis* to add the complete species name: “*Nematostella vectensis* Stephenson, 1935.” This could also be applied to the other species named in this work.

Response to reviewer: We have added the complete species name for anemones mentioned in the text.

Nematostella vectensis Stephenson, 1935

Scolanthus callimorphus Gosse, 1853

Actinia equina (Linnaeus, 1758)

Reviewer comment: Supplementary Figure 6: Can the image quality of this figure be improved? It seems to be relatively blurry compared to the other figures.

Response to reviewer: We have included an improved version of this figure in the revised manuscript.

Dominant toxin hypothesis: unravelling the venom phenotype across micro and macroevolution - Response to Reviewer #2

Reviewer comment: This is an interesting and clearly presented study of venom toxin evolution in sea anemones. It offers important new insights into patterns of venom variation within and between species, and the processes responsible for them. The authors used a battery of complementary techniques to generate these in-depth insights. I enjoyed reading this paper. I should point out that I lack firsthand experience in most of the techniques used in this study, so my technical evaluation should be considered that of an interested consumer, not an expert. I have only a few comments.

Response to reviewer: We would like to thank Reviewer 2 for their careful review of our manuscript. We are excited that they share our enthusiasm for the paper and have we have addressed all of their comments below.

Reviewer comment: As a general comment, this paper makes its inferences on the basis of venom transcriptomes. It is known mostly from other taxa (but also sea anemones: Madio et al., 2017, J. Proteomics 166) that transcriptomes are imperfect (sometimes very) predictors of proteomic venom profiles. I think that the authors should highlight this constraint of the study, and if possible if such data exists, outline why they think that these transcriptomic insights are likely to be reliable proxies for actual venom composition.

Response to reviewer: While the Madio et al. (2017) study represents a valuable contribution to the venom literature, we find it difficult to draw conclusions from this study regarding correlation between transcript and protein/peptide abundance in sea anemone venom. The reported lack of correlation is challenging to assess considering the different sample types (proteome = milked venom vs transcriptome = tentacles), the transcriptome sample was collected 72 hours after proteome sampling (obtained through electrostimulation - an acute stress treatment), and there was no biological or technical replication.

Nevertheless, we agree that protein/peptide expression levels may not always be accurately predicted by gene expression profiles and that this requires further justification in our manuscript.

“A potential constraint of this phylotranscriptomic approach is the assumption that toxin transcript abundances accurately represent the venom phenotype. The correlation between transcript abundances and protein/peptide expression has been the subject of debate within the venom field (Casewell et al., 2014; Jenner et al., 2019; Madio et al., 2017; Rokyta et al., 2015), and more widely (e.g., Li et al., 2014; Wang et al., 2019). Nevertheless, we consider our transcriptomic approach robust for the following three reasons: Firstly, a previous quantitative interspecies study did not find evidence of protein-level buffering in venoms that could complicate interspecific comparisons (Rokyta et al., 2015). Secondly, to avoid known issues with false positives in

transcriptomic analyses, we applied stringent filters to restrict our analyses to bona fide sea anemone toxins. Lastly, our work with N. vectensis (Columbus-Shenkar et al., 2018; this study) has shown strong congruence between toxin transcript and protein/peptide abundance.”

Reviewer comment: Page 3, 2nd paragraph: “high mutation rates of duplication”; should this be “high mutation rates of duplicated genes”?

Response to reviewer: We have amended the text as requested.

Reviewer comment: Page 5, 2nd paragraph: “macro-” instead of “macro”; “undergoes” instead of “undergo”. I will not identify other minor language errors. I leave it to the authors to make the necessary corrections throughout the manuscript.

Response to reviewer: We have made the recommended changes to the text. Furthermore, we have further proofread and corrected the manuscript for additional language errors.

Reviewer comment: Page 7, figure 1: Actinioidea, Edwardsioidea, and Metrioidea are superfamilies, not families. It would also be useful to non-expert readers to get a little background so they can appreciate how broadly these samples represent actiniarian diversity. Also please indicate for fig. 1b that the colours correspond to those of the toxins in fig. 1a.

Response to reviewer: We agree with the reviewer that it is important to highlight the taxonomic breadth of our sampling strategy and have included the following description to the text:

“We analyzed transcriptomes from 29 sea anemone species, spanning three of the five Actiniarian superfamilies (Actinioidea, Edwardsioidea, and Metrioidea).”

Further, we have altered the figure legend to clarify that the colors in Figure 1B correspond to the toxin families in Figure 1A.

“Model of best fit highlighted in color based on weighted AIC and are colored according to the toxin family key in panel A.”

Dominant toxin hypothesis: unravelling the venom phenotype across micro and macroevolution - Response to Reviewer #3

Reviewer comment:

What are the noteworthy results?

This well written manuscript highlights an emerging theory in venom research concerning gene expression and regulation of dominant toxin types in the venom arsenal. Here they identify the dominant toxin in sea anemones, Nv1, similar to the dominant snake toxins, three-finger toxins (TFTx), Phospholipase A2 (PLA2) and snake venom metalloprotease (SVMP). Using phylogenomic and transcriptomic tools the research team provides convincing evidence for their claim that "gene duplication-driven dominance by a single toxin family is a fundamental process shaping the venom phenotype."

Will the work be of significance to the field and related fields?

The work is significant to the venom field and follows on the heels of a recent finding in Snakes by Barua & colleagues (2019, 2021) where they use similar techniques to demonstrate that snake toxins are made up a few dominant families and gene regulation and duplication is the mechanism driving venom evolution.

This will have significant implications in related fields of evo-devo, molecular and cellular biology as at the core is the study of how complex traits evolved and the characterization of novelty in gene expression.

How does it compare to the established literature? If the work is not original, please provide relevant references.

The manuscript is well cited and reflects the established literature driving the field.

Response to reviewer: We would like to thank Reviewer 3 for their insightful comments on our manuscript. We are pleased that they found our manuscript provides convincing evidence for our central thesis, agree that this study has broad appeal across disciplines, and find that it reflects leading research in the field.

Reviewer comment:

Does the work support the conclusions and claims, or is additional evidence needed?

The weakest part of the manuscript is discussion on the genomic arrangement of the Nvi loci as part of the role of gene duplication in venom phenotype. This is very speculative and the evidence provided is not conclusive. I recommend shortening this section, specifically the discussion on the Florida halpotypes.

This can be significantly condensed.

Response to reviewer: We respectfully disagree with the reviewer that this section should be condensed as we feel that the often-overlooked mechanisms driving these expansions and contractions warrants discussion. We find that the organization of variants both within and between haplotypes is not random and offers valuable insight into the evolution of this locus and copy number variable loci in general. In particular, the deletion signatures surrounding the Florida haplotypes require discussion as these data do not support the most parsimonious hypothesis that they arose from a single deletion event.

Reviewer comment:

Are there any flaws in the data analysis, interpretation and conclusions?

The only flaw is stated by the authors as they have considered it. That is, in order to truly prove the omnigenic model for venom evolution as they are proposing they would need to do investigate the gene regulatory network involved in venom production by doing both computational analyses (using comparative transcriptomics of nematocytes and gland cells) followed by in vivo genetic manipulation of the gene regulatory network to confirm the computational findings. Currently there are few in vivo model systems for conducting this work, anemones are among the few, but this is beyond the scope of this publication. It is where the venom field has to go to support these claims about dominant toxin theories. If you knock out the dominant toxin gene, what are the implications for the venom arsenal and it's function and can this be inherited for several generations?

Response to reviewer: We wholeheartedly agree with the reviewer that, while beyond the scope of this study, genetic manipulation is the direction that the venom field needs to go and that *N. vectensis* offers an exciting model system to functionally test dominant toxin theories.

Reviewer comment: **Another minor flaw is that there are no images of the anemones used in the study. Adding representative images of the three families used in Figure 1 would go a long way in terms of science communication and grabbing attention of the reader.**

Response to reviewer: Following the excellent suggestion by the reviewer we have added three representative pictures for Figure 1, representing each of the sea anemone superfamilies.

Reviewer comment:

Do these prohibit publication or require revision?

No.

Is the methodology sound? Does the work meet the expected standards in your field?

Yes.

Is there enough detail provided in the methods for the work to be reproduced?

Yes.

Response to reviewer: We are pleased that the reviewer finds our manuscript suitable for publication and that our methods are sound and described in sufficient detail.

Reviewers' Comments:

Reviewer #1:

Remarks to the Author:

The authors addressed all of the comments from the previous review and I find the manuscript much improved.

Reviewer #2:

Remarks to the Author:

The authors have addressed my comments. I look forward to seeing the paper published.

Reviewer #3:

None